

# Resounding failure to replicate links between developmental language disorder and cerebral lateralisation

Alexander C. Wilson and Dorothy V.M. Bishop

Department of Experimental Psychology, University of Oxford, Oxford, United Kingdom

## ABSTRACT

**Background**. It has been suggested that failure to establish cerebral lateralisation may be related to developmental language disorder (DLD). There has been weak support for any link with handedness, but more consistent reports of associations with functional brain lateralisation for language. The consistency of lateralisation across different functions may also be important. We aimed to replicate previous findings of an association between DLD and reduced laterality on a quantitative measure of hand preference (reaching across the midline) and on language laterality assessed using functional transcranial Doppler ultrasound (fTCD).

**Methods**. From a sample of twin children aged from 6;0 to 11;11 years, we identified 107 cases of DLD and 156 typically-developing comparison cases for whom we had useable data from fTCD yielding a laterality index (LI) for language function during an animation description task. Handedness data were also available for these children.

**Results**. Indices of handedness and language laterality for this twin sample were similar to those previously reported for single-born children. There were no differences between the DLD and TD groups on measures of handedness or language lateralisation, or on a categorical measure of consistency of left hemisphere dominance. Contrary to prediction, there was a greater incidence of right lateralisation for language in the TD group (19.90%) than the DLD group (9.30%), confirming that atypical laterality is not inconsistent with typical language development. We also failed to replicate associations between language laterality and language test scores.

**Discussion and Conclusions**. Given the large sample studied here and the range of measures, we suggest that previous reports of atypical manual or language lateralisation in DLD may have been false positives.

Corresponding author
Alexander C. Wilson,
alexander.wilson@psy.ox.ac.uk,
alexander.wilson2@psy.ox.ac.uk

## INTRODUCTION

The relationship between atypical brain lateralisation and developmental language disorder (DLD) has intrigued scientists for many years, but is still not well understood. Lateralisation is thought to reflect an adaptive process of specialization by which cognitive functions become preferentially supported by one cerebral hemisphere: in the case of language, typically the left hemisphere. Theoretical accounts have suggested that individuals who do not show the population bias towards left hemisphere dominance for language may be at risk of disrupted language development (e.g., *Annett, 2002*; *Bishop, 2013*; *Crow et al., 1998*).

The earliest studies to test the relationship between hemispheric dominance and language problems measured laterality by handedness. Handedness has been used as a marker of language lateralisation given a link between the two traits: left-handers are more likely than right-handers to have atypical lateralisation for language (*Knecht et al., 2000*; *Szaflarski et al., 2002*). If atypical laterality is associated with language and literacy disorders, then the expectation is that more left-handers will have these disorders. While handedness assessed via a questionnaire inventory has shown no link with speech and language problems (*Bishop, 2001*; *Bishop, 2005*), a relationship has been reported for reduced right hand preference in a task requiring reaches across the midline (*Bishop, 2005*; *Hill & Bishop, 1998*). With respect to literacy, a meta-analysis indicated a significant over-representation of left-handers among those with dyslexia (*Eglinton & Annett, 1994*). However, this meta-analysis did not weight the effects of individual studies for their sample size and did not calculate a summary effect size, so the reported relationship is difficult to interpret. Importantly, there are also numerous large epidemiological studies failing to find any evidence of a link between handedness and reading problems (e.g., *Levinson, 1988*; *Rutter & Yule, 1970*; *Satz & Fletcher, 1987*). Investigation of handedness as a predictor of language ability in typically developing individuals provides further uncertain (probably null) results. For instance, a meta-analysis found no relationship in the full analysis ($N = 359{,}890$), and a very small disadvantage for left-handers (Hedge's $g = -0.09$) when only children were analysed (*Somers et al., 2015c*). Methodologically, there is concern in the field that flexible criteria for categorisation of handedness, as well as selective reporting of results *only* when a significant effect is found, may have led to inflated type 1 error (*Bishop, 1990*). Overall, the evidence provides weak grounds for predicting that DLD is related to reduced right handedness.

It is important to note, however, that manual laterality is at best a weak proxy for language lateralisation in the brain (*Groen et al., 2013*). A smaller literature using more direct brain measures of language laterality does appear to support the view that there may be reduced left hemisphere dominance in those with DLD. Small-scale studies using functional transcranial Doppler sonography (fTCD) have compared task-related blood flow in the middle cerebral arteries (MCAs) during productive language paradigms. *Illingworth & Bishop (2009)* found reduced left lateralisation in a sample of dyslexic adults ($n = 30$) compared to controls, while *Bishop et al. (2014)* reported that four year-olds with language problems ($n = 11$) did not have significantly left-lateralised language function at the group level, whereas those with typically developing language showed the usual left bias found in adults. Compared to typical controls, *Whitehouse & Bishop (2008)* indicated that a pattern of right and bilateral distribution of language function characterised adults with persisting specific language impairment ($n = 11$), whereas typical laterality was found for those whose language problems had resolved ($n = 9$) and a group with autism spectrum disorder ($n = 11$). This laterality difference between those with autism and those with language difficulties chimes with *Lindell & Hudry (2013)* literature review of language lateralisation in autism, which provided the strongest evidence for atypical laterality in individuals with ASD who *also* had comorbid language difficulties. This supports the view that atypical laterality is relevant to other neurodevelopmental disorders but particularly implicated in DLD.

Functional MRI studies, in which blood oxygenation levels during language tasks are compared to a baseline, corroborate the fTCD findings reported above. Thus, reduced left laterality was found in two small samples of children with specific language impairment ($n = 21$; $n = 10$) compared to controls (*Badcock et al., 2012a*; *De Guibert et al., 2011*). A further study reported greater right lateralisation in children with speech delay ($n = 17$) compared to controls (*Bernal & Altman, 2003*), though statistical significance was only reached in the latter study when decomposing the sample by age. Laterality indices from fMRI studies of individuals with dyslexia have also indicated reduced left laterality (*Waldie et al., 2013*; *Xu et al., 2015*), though the clinical samples again were small ($n = 12$ in both). In addition, structural MRI studies provide some evidence for atypical structural asymmetries, especially in frontal language regions, in children with language disorder, as reviewed by *Mayes, Reilly & Morgan (2015)*. This review presented limited evidence for the hypothesis that typical cerebral asymmetries are disrupted in language disorder.

Research assessing whether cerebral laterality predicts language skills in typically developing individuals provides some further evidence that left hemisphere dominance is advantageous. A moderate positive relationship has been reported between left lateralisation and vocabulary and non-word reading skills using fTCD ($n = 55$) (*Groen et al., 2012*); a trend with word reading did not meet significance. Meanwhile, *Everts et al. (2009)* found a large correlation ($r = .59$) between verbal IQ and left lateralisation in a fMRI vowel detection task in a sample of 20 adolescents; the correlation with laterality derived from a fMRI synonym decision task was moderate in size but not statistically significant. In 24 young adults, left lateralisation of Wernicke's area during fMRI productive language tasks was part of a principal component also including greater functional connectivity at rest and greater symmetry of the arcuate fasciculus that predicted verbal IQ ($r = 70$) (*Piervincenzi et al., 2016*). An interesting counterpoint to these findings is an fMRI study of language lateralisation in a sample of patients with callosal agenesis ($n = 25$) (*Hinkley et al., 2016*). While there was no correlation between laterality and verbal IQ in 21 healthy matched controls, there was a high correlation in the patients ($r = .55$). The lack of relationship in the healthy controls contradicts other findings, though the restricted variance in the group may explain this. However, the effect reported for the patients suggests that where normal lateralisation processes are disrupted, the recruitment of the left hemisphere for language is most adaptive for language development.

However, results have not been unambiguous in their support of the hypothesis that DLD is associated with atypical language lateralisation. For instance, *Berl et al. (2014)* found non-significant correlations between several language measures and left lateralisation of Wernicke's area in a sample of 4–12 year-olds ($n = 56$), contrary to the effects reported in other studies listed above—though there was a significant correlation with lateralisation of the cerebellum. Counterintuitively, right-lateralised language-related activity in the cerebral cortex has not always been reported as detrimental for language development. Thus, *Van Ettinger-Veenstra et al. (2010)* reported a relationship between more right lateralised language activity and better performance on neuropsychological tests of language and reading ability ($r = $ around $-.5$) in 14 healthy adults. In a large study that oversampled left handers (153 in a total sample of 297), individuals with strong hemispheric dominance

for language (whether left or right) showed slightly stronger performance than those with more symmetrical language laterality on verbal, spatial and verbal memory components of a cognitive battery, though the effect was very small ($\eta 2 = 0.03$) (*Mellet et al., 2014*). Thus, evidence is mixed, but we can infer from this unusually large fMRI study that atypical laterality does not necessarily entail a cognitive disadvantage. Some atypically lateralised individuals clearly perform above average on verbal and non-verbal assessments.

Given this lack of a simple link between laterality and language skills, it is possible that a more complex relationship exists between the two. For one thing, it need not be assumed that all language functions show the same pattern of lateralisation within the individual, especially given that several networks seem to be implicated in different aspects of language processing (*Friederici, 2011*). *Bishop (2013)* proposed that the lateralisation of different aspects of language processing may show one of two endophenotypes in the individual: a left-brain bias that promotes left-hemisphere mediation of all language functions vs an unbiased brain, where there is equal likelihood of different language functions developing in the left or right hemisphere. Given the partial relationship between language laterality and handedness (e.g., *Knecht et al., 2000*), these endophenotypes may also apply to manual laterality, with the left-brain bias promoting left hemisphere dominance for motor functions and the unbiased brain leaving handedness to chance. This model is in line with earlier genetic models in which the recessive allele of a hypothetical "laterality gene" was thought to remove the typical bias for both left language lateralisation and right-handedness (*McManus, 1985*; *Annett, 2002*). Current research rejects the hypothesis that language lateralisation and handedness are controlled by a single gene (*Ocklenburg et al., 2014*), but there is evidence that the "no bias" endophenotype for handedness predicts a similar "no bias" endophenotype for language lateralisation, as these earlier theories proposed. This evidence includes two fMRI studies of monozygotic twins that found significant concordance for language lateralisation only within twin pairs concordant for handedness (*Badzakova-Trajkov, Häberling & Corballis, 2010*; *Sommer et al., 2002*). Twins discordant on the laterality measures provide evidence for the existence of a "no bias" endophenotype in which laterality develops randomly within twins and genetic control over laterality is lost. The presence of this "no bias" endophenotype predicts that overall heritability of laterality across the population should only be modest, and empirical data support this. In a study of 368 people from 37 families, handedness heritability was 0.24 and language lateralization by fTCD was 0.31—and it is worth noting that this may be an overestimation given that families were oversampled for cross-generational sinistrality (*Somers et al., 2015b*).

The left-brain bias model assumes that the bias operates separately and in a probabilistic fashion for different functions. A person with left-brain bias will tend to have all language-related functions mediated preferentially by the left hemisphere, and is likely to show right handedness. A person with no bias is more likely to have discrepant lateralisation for different functions, and this may increase the risk of developmental language problems. According to this model, a single measure of lateralisation will give only a crude indication of whether a person is in the left-bias category. However, where individuals show consistent left laterality on different measures of language laterality, and potentially on measures of handedness too, this is likely to indicate that they are of the left-bias endophenotype, which

may be protective against language problems. Based on this model and existing research on laterality and DLD, we hypothesise (a) that reduced left lateralisation is associated with DLD, and (b) that discrepancies in laterality across measures of language lateralisation and handedness is a risk factor for DLD.

In the current study, we aimed to test the left-brain bias model, using data from a sample of twin children who had been assessed on language and literacy skills, as well as on two measures of handedness and a direct measure of cerebral lateralisation for language. The sample had been selected to be over-representative of cases of DLD. In the current paper, the relationship between language and laterality is probed, with the twin status of the children taken into account using multi-level modelling. In a related paper, we consider heritability of laterality assessed by comparing monozygotic and dizygotic twins.

## METHODS

We report how we determined our sample size, all data exclusions, all manipulations, and all measures in the study.

### Participants

We recruited families with twin children aged between 6;0 and 11;11 years, whose first language at home was English. We aimed for an over-representation of twin pairs in which one or both twins had language or literacy problems that might be indicative of DLD. Families were recruited via fliers sent to primary schools around the UK, advertisements on our group's website and via twins' clubs. The initial flier was worded as follows: 'We are looking for sets of twins to participate in a new study investigating factors underlying children's language difficulties. We want to test twins with and without language problems (language-impaired, typically-developing, or one twin of each)'. Head teachers were asked to forward information sheets about the study to parents of twin children. We aimed to recruit 180 pairs selected on the basis of having language or literacy problems (60 MZ, 60 DZ opposite sex and 60 DZ same sex), and 60 unselected pairs (20 of each type). In practice, self-selection of those volunteering to take part meant that the latter group tended to come from relatively highly educated backgrounds, and could not be regarded as representative of the general population. The flow chart in Fig. 1 shows the numbers of participant children at different stages of selection. Parents and caregivers of 194 twin pairs volunteered for the study, yielding 134 children who met our criteria for DLD, and 190 children who met criteria as typically developing (TD). See the Data Analysis section below for criteria.

Children were excluded from the sample if they met any of the following criteria: WASI nonverbal ability (performance IQ) more than two SDs below the population mean; diagnosis of autism spectrum disorder (ASD) in one or both twins; sensorineural hearing loss or failure of a hearing test on the day of testing; and brain injury or a serious medical condition affecting one or both twins. In order to test our main hypothesis that DLD was related to cerebral laterality as measured by fTCD, it was necessary to exclude individuals in a second stage of exclusions if we did not obtain useable fTCD data from them, defined as fewer than 12 accepted trials. We also excluded participants with extreme laterality indices

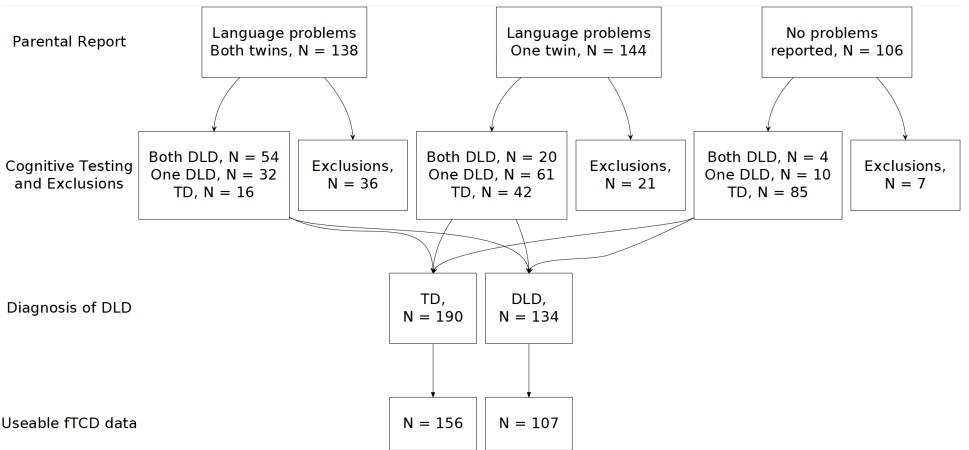

**Figure 1  Chart showing the flow of participants through the study.** Thirty-five children were excluded because they or their twin were reported as having an ASD diagnosis; four were excluded because their IQ was below 70; 12 were excluded because they failed the hearing test; and 13 were excluded for other reasons, such as a medical diagnosis.

(above 10 or below −10); 3 individuals were excluded based on this criterion. Useable fTCD data were obtained from 107 (80%) of the children with DLD, and 156 (82%) of the TD children.

## MATERIAL

### Language, literacy and cognitive assessments

The assessment battery used to categorise language status is shown in Table 1.

Nonverbal ability was estimated using the two nonverbal subtests of the WASI (Block Design and Matrices); age-normed scores on these were converted into an overall Performance IQ score. The remaining tests of the battery were used to index language and literacy abilities. All measures involved individual assessment by a trained examiner, except for the CCC-2, which is a parental report instrument.

### Laterality assessments
#### Handedness

Handedness was assessed using the same hand preference battery as in *Bishop (2005)*. This was based on items from the Edinburgh Handedness Inventory (EHI) (*Oldfield, 1971*), modified to replace one item (striking a match) deemed unsuitable for children. The experimenter asked the child to demonstrate how they would perform each of the following actions: writing, drawing, throwing a ball, using scissors, using toothbrush, cutting with a knife, using a spoon, using a broom, taking the lid off a box, and dealing cards. One point was awarded for exclusive right hand use, zero points for left hand use, and half a point if both hands were used, giving a score ranging from zero to ten. This score was then converted to a laterality index ranging from −100 (extreme left handedness) to 100 (extreme right handedness). In addition to this score, a child was categorised as right-handed if they scored above 0 on this measure.

**Table 1  Assessment battery.**

| Instrument | Measure |
| --- | --- |
| Wechsler Abbreviated Scale of Intelligence (WASI) (*Wechsler, 1999*) | Block design |
| | Matrices |
| | Vocabulary |
| Woodcock Johnson III tests of cognitive abilities (*Woodcock, McGrew & Mather, 2007*) | Verbal comprehension |
| NEPSY: a developmental neuropsychological assessment (*Korkman, Kirk & Kemp, 1998*) | Sentence repetition |
| | Repetition of nonsense words |
| | Oromotor sequences |
| Phonological Assessment Battery (PhAB) (*Frederickson, Frith & Reason, 1997*) | Picture naming test |
| | Digit naming test |
| Test of Word Reading Efficiency (TOWRE) (*Torgesen, Wagner & Rashotte, 1999*) | Sight word efficiency |
| | Phonetic decoding efficiency |
| Neale analysis of reading ability—2nd British edition (NARA-II) (*Neale, 1997*) | Reading accuracy |
| | Reading comprehension |
| | Reading rate |
| Children's Communication Checklist-2 (CCC-2) (*Bishop, 2003*) | General communication composite |

### Quantitative hand preference

A measure of strength of hand preference was obtained from the second measure of handedness, the Quantification of Hand Preference (QHP) task (*Bishop et al., 1996*). This measures an individual's tendency to continue to use the preferred hand when items are placed across the midline. In this task, the child stands in front of a semi-circular array of picture cards, with three cards in each of seven positions extending at 30-degree intervals from the left to the right of the child's midline. The child is asked to pick up a named card and place it in a central box. The child is not told that handedness is being assessed, and no instructions are given about how to handle the cards or how to stand, other than that to remain in the central location in front of the box. The same quasi-random order of positions is used for all children, starting with a card at the midline and continuing until the child has reached for three cards at each of seven locations. One point was recorded for each right-handed reach, giving a possible total of 21. In addition to this quantitative score, a child was categorised as right-handed by this measure if they scored over 10 points.

### Language laterality

Language laterality was assessed using functional transcranial Doppler ultrasound (fTCD) while the child performed a productive language task. Transcranial Doppler ultrasound is a technique used in medical contexts to assess the integrity of the cerebral blood vessels using ultrasound probes placed on the temples. In this study, probes were attached to a headset

and positioned to detect blood flow in the left and right middle cerebral arteries (MCAs), which supply language-relevant regions in the lateral aspects of the frontal, temporal and parietal lobes. Researchers conducting the procedure were trained to identify the blood vessels, which have distinctive characteristics in terms of depth and direction of flow.

The language task used was the animation description paradigm, for which a video demonstration can be accessed from *Bishop, Badcock & Holt (2010)*, and which was implemented with children by *Groen et al. (2012)*. On each trial, the child silently views a 12 s clip from a cartoon including sounds but no speech. A response cue then indicates the start of a 10 s talk phase during which the child is asked to describe what happened in the cartoon. A second cue then indicates that the child should stop talking. This paradigm has previously been found to have good validity and reliability (*Bishop, Watt & Papadatou-Pastou, 2009*). A maximum of 30 trials was administered, depending on the child's tolerance of the procedure. The child's verbal responses were recorded and subsequently transcribed, and the examiner noted behaviour during the procedure. Trials were excluded where the child either spoke during a silent period, or failed to talk during the talk phase: these infringements need to be omitted because they invalidate analysis of the trial, which involves comparing cerebral blood flow during the period of interest when the child talks with a baseline period when no talking occurs. The baseline is taken to be the 12 s spent watching the animation immediately before the talk phase and the period of interest commences 4 s into the talk phase and lasts for 10 s. The 4 s lag allows time for cerebral blood flow changes associated with speech to take place.

The analysis of the animation task data consists of a standard sequence of processing steps, following work by *Deppe et al. (1997)*, including removal of the heartbeat (heart cycle integration), signal normalisation, artefact rejection, epoching and baseline correction. Following this initial processing, a Laterality Index (LI) was calculated for each individual. First, grand mean curves for blood flow velocity in each MCA were produced by averaging across all valid trials at each time point during the course of a trial. Then, the time point reflecting the greatest difference in velocity between the two MCAs during the period of interest was identified, and a 2 s window centred on that point was constructed. The mean velocity in the left MCA during this window minus the mean in the right MCA gave the LI.

In previous studies, we have used Matlab to perform this analysis using the DopOSCCI toolbox (*Badcock et al., 2012b*). However, our research group is now moving to using R across the whole pipeline of data processing, so we created an R script to perform the same sequence of operations, while also analysing all files using DopOSCCI for comparison. The R script incorporated one additional feature, which enabled the identification of single data points during trials where there had been very brief signal dropout (typically due to movement of the probe) for substitution by the mean value of that trial. This avoids losing the whole trial because of one aberrant data point, while also correcting for an extreme value that could affect the signal normalisation procedure. Following *Groen et al. (2012)* we excluded data from children who had fewer than 12 accepted trials, as the LI is likely to be unreliable when based on such a small amount of data.

In addition to computing LIs, we also assigned each participant to a laterality category. For this purpose, we tested whether the LI was significantly different from zero, with the

*p*-value set to .05. To enable significance-testing, the standard error of each participant's LI was calculated thus: firstly, trial-by-trial LIs were computed for every valid trial during the 2 s window from which the participant's overall LI was derived, and then the standard deviation of these trial-by-trial LIs was divided by the square root of the number of valid trials to give the standard error of the overall LI. Next, the standard error was multiplied by $\pm 1.96$ and added to the LI, giving 95% confidence intervals around the LI. If these CIs did not cross with zero, laterality was classified as left or right depending on direction, and bilateral where the CIs did cross zero. Note that there is always a concern that a coding of bilateral laterality could result if data were merely noisy.

In addition to overall LI and laterality category, several other measures were taken. Firstly, LIs were derived separately from odd and even trials to allow computation of split half reliability. Secondly, we calculated the difference in blood flow between the mean for the period of interest relative to the mean of the immediately preceding baseline, averaged across all valid trials; this was done separately for the left and right MCAs. Since blood flow velocity is normalized to a mean of 100 and baseline-corrected during signal processing, a positive value indicates a percent increase during the period of interest, and a negative value a percent decrease. Finally, the mean number of words spoken by the child during valid trials was recorded.

### Left hemisphere dominance

According to the left-brain bias model (*Bishop, 2013*), if an individual shows consistent left hemisphere dominance across different laterality measures, it is likely to indicate that they belong to the left-bias category. On the other hand, inconsistency across measures would indicate that an individual is of the no-bias category, which may be a risk factor for DLD. To test this notion, children were identified as consistently left dominant if they were categorised as left lateralised for language by fTCD, and if they were categorised as right-handed on both handedness tasks. In all other cases, children were classed as not having consistent left hemisphere dominance.

## Procedure

Ethical approval was obtained for the study in 2011 from the Berkshire NHS Research Ethics Committee (reference 11/SC/0096), and data collection started in August of that year, finishing in October 2016. Where families had expressed interest in the study, they were interviewed by telephone to assess whether the children were likely to meet inclusion criteria, and if so, an appointment was made to see the twins at home or at school, depending on parental preference. Written consent was obtained from a parent/caregiver for their child's participation, and children signed a simplified assent form. Families were widely dispersed around the UK, including Northern Ireland, Scotland, Wales and England, so testing was scheduled where possible to minimise travel. During the course of recruitment, which lasted for a period of five years, a total of eight research assistants as well as the senior author were involved in assessing children. In some cases, two testers worked together, each seeing one twin, and in others, a single tester saw both children sequentially. The assessment was conducted in a single session lasting between 2 and 3 h per child, with breaks where needed.

**Table 2** Study variables.

| Independent variable | Dependent variables |
| --- | --- |
| Group (DLD or TD) | Handedness on the Edinburgh Handedness Inventory (EHI) |
| | Quantified Hand Preference (QHP) |
| | fTCD laterality index |
| | fTCD mean % change in left MCA blood flow during speech compared to baseline |
| | fTCD mean % change in right MCA blood flow during speech compared to baseline |
| | $N$ words produced per valid trial on fTCD |
| | Left hemisphere dominance across laterality measures |

## Data analysis

Study data were analysed using R software (*R Core Team, 2016*), with the main database managed using REDCap electronic data capture tools hosted at the University of Oxford (*Harris et al., 2009*). Original data are available on Open Science Framework (https://osf.io/ksqrf/?view_only=54f013aaf65d45a5924748179538756d).

Results from the language/cognitive test battery were used to categorise children as having DLD if they scored more than 1 SD below population norms on two or more out of 13 language/literacy measures, and as TD if they scored below this threshold on no more than one measure. In previous studies, we have excluded children who met this criterion solely on literacy measures (TOWRE and NARA-II); in the current sample, 10 children fell in this category and were included as DLD, on the basis that prior research has found atypical laterality in adults with dyslexia (*Illingworth & Bishop, 2009*). The mean number of tests on which a child with DLD underperformed was 4.21 (SD = 2.49). It should also be born in mind that the children with DLD had lower nonverbal ability than the TD children. In a multilevel model with performance IQ as dependent variable, group (TD or DLD) as a fixed factor and twin pair membership as a random factor, group showed a significant effect, $t(254.83) = 5.68$, $p < .001$. Marginal mean Performance IQ for the TD children was 106.95 (SD = 14.21) and for the children with DLD was 97.89 (SD = 13.81). Cohen's $d$ of the between-groups difference was 0.65. Notably, the IQ difference in this sample was largely down to the TD children "overperforming"; the mean for the children with DLD is close to the assumed population mean (i.e., 100).

To test our main hypothesis that DLD would be associated with reduced laterality, we made between-groups comparisons of means of quantitative variables or proportions of categorical variables. Table 2 shows the full set of independent and dependent variables considered in the analysis. Since participants were twins, it was necessary to account for the lack of independence of observations (*Kenny, Kashy & Cook, 2006*), and so we adopted a multilevel modelling approach analogous to that used by *Brookman et al. (2013)*.

To test whether mean fTCD LI differed between the DLD and TD groups, we used a multilevel model that considered group as a fixed effect and twin pair membership as a random effect. This model was run using the lme4 package (*Bates et al., 2015*) to specify the model, with the lmerTest package used for significance testing and generating estimated

marginal means (*Kuznetsova, Brockhoff & Christensen, 2016*). We also carried out this same analysis for the mean percent change in blood flow during the period of interest relative to the baseline in the left MCA and in the right MCA separately. A multinomial multilevel model, with categorisation of fTCD laterality as left, bilateral or right as the dependent variable, was used to test whether individuals with DLD were more likely than TD individuals to show right or bilateral compared to left laterality. The model estimated two logit equations, each comparing left laterality to one of the atypical lateralities, and assigned predicted log-odds to each comparison. We used the MCMCglmm package in R to run the model (*Hadfield, 2010*). As the package adopts a Bayesian approach (Markov Chain Monte-Carlo iterative sampling), 95% credible intervals were calculated around the predicted log-odds, and we took no overlap with zero to indicate significance. Predicted log odds are reported as odds ratios for ease of interpretation. We also plot the grand average curves for both groups showing change in flow in the two MCAs and change in laterality over the course of a trial.

We assessed the ability of group to predict the quantitative handedness measures (the QHP and the adapted EHI) using inflated beta regressions. These were implemented with the GAMLSS package (*Stasinopoulos & Rigby, 2007*). For analysis, scores were rescaled to range between 0 and 1. As the handedness measures are bounded and skewed, they approximate a beta distribution; this can be modelled optimally using a beta regression (*Ferrari & Cribari-Neto, 2004*). While zeros and ones are not possible within a beta distribution, these can be scored on the handedness measures (i.e., extreme left and extreme right handedness), and therefore we used inflated beta regression, which can incorporate these values through a mixture model (*Ospina & Ferrari, 2012*). To our knowledge, existing packages in R do not allow the inclusion of a random effect in inflated beta regression. Therefore, instead of modelling twin pair membership as a random effect, we ran the regressions in two replication samples with twin one in one sample and twin two in the other, in order to deal with the non-independence of observations. Twins were arbitrarily labelled as such at the start of the study, so these qualify as random samples. All models included the logit function of the handedness measure (either the adapted EHI or QHP) as the dependent variable and group as predictor. Beta coefficients and associated *p*-values are reported for group, and we required $p < .05$ in both samples for a significant effect of group on that particular measure. In an attempt to replicate the previously reported relationship between language disorder and a reduced tendency to reach across the midline (*Bishop, 2005*; *Hill & Bishop, 1998*), we also assessed whether the probability of reaching to the seven spatial positions in the QHP task differed between the DLD and TD groups. A multilevel model was applied to the data using the lme4 package, with twin pair membership as a random effect, spatial position as a within factor and group as a between factor. Main effects and the interaction are reported.

To complete the main analysis, we tested the prediction that individuals with DLD would show less evidence of consistent left hemisphere dominance on the three laterality measures used in this study. An individual was coded as left hemisphere dominant if they were left lateralised on fTCD, and were right-handed on both the adapted EHI and QHP; otherwise, they were coded as not consistently left hemisphere dominant. In testing the

hypothesis, we ran a multilevel binary logistic regression using the R package mse4, with left hemisphere dominance as the fixed effect and twin pair membership as a random effect. We report significance testing of the fixed effect and the odds ratio with profile likelihood 95% CIs that a TD individual would be consistently left hemisphere dominant compared to a DLD individual being consistently left hemisphere dominant.

A subsidiary aim was to see whether we could replicate associations between language laterality and language tests found by *Groen et al. (2012)*. Analysis was conducted in two replication samples, as described above, so that data from only one twin of each twin pair contributed to each analysis. Spearman's method was used, since the laterality index is not normally distributed, and all 13 language measures as well as performance IQ were included. We applied the Holm-Bonferroni correction to each of the two sets of correlations to adjust for multiple testing, and required a correlation to be significant in both samples for it to be classed as a true effect.

## RESULTS

### Preliminary Analysis of fTCD language laterality

Table 3 shows descriptive statistics for all measures; the language measures are reported as standard scores. In our sample, fTCD results indicated that 61.50% of the typically developing children were left lateralised for language in the animation description task; 18.60% were categorised as bilateral and 19.90% as right lateralised. Respective percentages for the children with DLD were 72.90% as left, 17.80% as bilateral and 9.30% as right.

We checked the reliability of the fTCD LI by computing the correlation between the LIs calculated separately for even and odd trials. Split-half reliability was excellent at the full sample level, $r = .84$, and when dividing participants into the TD children, $r = .86$, and those with DLD, $r = .78$. Number of words spoken during valid trials did not predict laterality index, $r = .03$, $p = .614$, indicating that any differences in laterality detected by fTCD cannot be attributed to quantity of speech produced. This impression is supported by a strikingly similar mean number of words produced by left-lateralised children (M = 20.06, SD = 4.86) compared to bilateral (M = 19.43, SD = 5.38) and right-lateralised children (M = 19.87, SD = 5.38). We also checked whether the two groups differed in terms of the number of fTCD trials included in analysis. There was a significant difference, $t(203.37) = 2.62$, $p = .009$, with more trials available for the TD children (M = 26.78, SD = 3.56) than the children with DLD (M = 25.48, SD = 4.18). The effect size was small (Cohen's $d = 0.33$). Finally, we checked for any sex and age differences in LI using a multilevel model with twin pair membership as a random effect and sex and age as fixed effects. Age showed no effect, $t(147.58) = 0.50$, $p = .617$, but sex did, $t(198.45) = -2.24$, $p = .026$. Marginal mean LI for boys was 2.10 (SD = 2.78) and for girls was 1.32 (SD = 2.78). The effect size was small (Cohen's $d = 0.28$). As we did not make any prediction about sex in this study, we did not include the effect of sex in our hypothesis-testing models. However, seeing as it showed a relationship with fTCD laterality, we do report some exploratory analysis of sex by group interactions in Supplemental Information.

**Table 3  Descriptive results for all variables.** Means (and SDs) are presented for continuous variables, and frequencies (and percentages) are presented for categoric variables.

|  | TD | DLD |
| --- | --- | --- |
| **Sample characteristics** | | |
| N, children | 156[a] | 107[b] |
| Age, years | 9.1 (±1.6) | 8.9 (±1.5) |
| Gender, male | 56 (35.9%) | 69 (64.5%) |
| **Cognitive measures** | | |
| Performance IQ | 107.6 (±13.5) | 96.6 (±12.4) |
| Vocabulary | 57.5 (±8.6) | 45.3 (±9.7) |
| Verbal comprehension | 105.4 (±8.8) | 97.3 (±9.3) |
| Sentence repetition | 10.2 (±2.7) | 6.8 (±2.9) |
| Repetition of nonsense words | 11.6 (±1.9) | 9.2 (±2.7) |
| Oromotor sequences | 3.2 (±1.0) | 1.9 (±0.9) |
| Picture naming test | 109.2 (±13.1) | 92.1 (±15.8) |
| Digit naming test | 109.8 (±12.6) | 94.1 (±17.1) |
| Sight word efficiency | 113.2 (±11.1) | 93.7 (±17.2) |
| Phonetic decoding efficiency | 111.6 (±13.1) | 93.9 (±14.0) |
| Reading accuracy | 107.0 (±10.2) | 89.5 (±11.7) |
| Reading comprehension | 106.4 (±9.5) | 88.8 (±10.1) |
| Reading rate | 108.3 (±10.3) | 94.9 (±14.9) |
| General communication composite | 86.5 (±15.6) | 62.8 (±22.1) |
| **Laterality measures** | | |
| Edinburgh Handedness Inventory (EHI) | 64.0 (±59.0) | 68.0 (±52.5) |
| N children right-handed by EHI | 134 (85.9%) | 93 (86.9%) |
| Quantified Hand Preference (QHP) | 14.3 (±7.5) | 15.4 (±6.2) |
| N children right-handed by QHP | 114 (73.1%) | 86 (80.4%) |
| fTCD laterality index | 1.5 (±3.0) | 2.0 (±2.4) |
| Mean % change in left MCA blood flow | −1.0 (±4.8) | 0.5 (±3.9) |
| Mean % change in right MCA blood flow | −2.2 (±4.9) | −1.0 (±3.4) |
| N words produced per trial on fTCD | 20.7 (±5.1) | 18.7 (±4.7) |
| N children with consistent left hemisphere dominance | 70 (44.9%) | 58 (54.2%) |

**Notes.**

[a]*Measures without complete data for all the TD children*: Picture Naming Test, N = 154; Digit Naming Test, N = 155; Reading Accuracy, N = 150; Reading Comprehension, N = 150; Reading Rate, N = 150; General Communication Composite, N = 136; N words produced per trial on fTCD (since recording was not available for all children), N = 152.

[b]*Measures without complete data for all the children with DLD*: IQ, N = 106 Picture Naming Test, N = 106; Sight Word Efficiency, N = 106; Phonetic Decoding Efficiency, N = 102; Reading Accuracy, N = 98; Reading Comprehension, N = 97; Reading Rate, N = 98; General Communication Composite, N = 81; N words produced per trial on fTCD (since recording was not available for all children), N = 103.

## Comparison with single-born sample

It has been suggested that twinning is a factor affecting development of laterality due to twin-specific phenomena, such as mirror-imaging (*Newman, 1928*), hormonal transfer (*Elkadi, Nicholls & Clode, 1999*) and birth order (*James & Orlebeke, 2002*), and due also to an increased general risk of perinatal complications. Early adversity affecting the left hemisphere may be associated with the compensatory development of atypical right hemisphere dominance, promoting "pathological left-handedness" (*Annett, 1985*),

although this theory was not supported in a large longitudinal data-set that found no relationship between atypical handedness and birth stress (*McManus, 1981*). *Medland et al. (2009)* presented evidence from 54,270 twins and their siblings for an absence of any twin-specific effects: prevalence of left-handedness did not differ between twins by zygosity, sex or birth order, and nor did it differ between twins and their non-twin siblings.

While the empirical findings speak against twin-specific effects, we nonetheless considered whether laterality was unusual in the typically developing twins compared to single-born children, before proceeding to the main analysis. We compared the TD children in this sample to a previous sample of single-born children ($N = 58$) tested using the same task with fTCD (*Groen et al., 2012*). Although more of the TD twins show right-sided laterality than the children in the previous sample, overlapping 95% high density intervals between the two groups indicate no difference in central tendency. See the pirate plot below in Fig. 2 for the distributions of LIs. Handedness, as measured by the adapted EHI was also very similar among the TD twins and the single-born sample (twins, M = 65.67, SD = 56.37; single-born children, M = 63.57, SD = 44.53).

The twin-specific effects outlined above are predicted to influence laterality based on zygosity. Mirror-imaging, for instance, is thought to increase the risk of atypical laterality in MZ twins if the embryo does not split until after the left–right axis forms (*Newman, 1928*). In testing for differences in laterality index by zygosity in a multilevel model with twin pair membership as a random effect, zygosity was not significant when included as a fixed effect, $t(137.14) = 0.82$, $p = .414$. Marginal mean laterality index [95% CIs] was 1.83 [1.32–2.34] for MZ twins, 1.59 [1.01–2.18] for DZ same sex twins and 1.45 [0.68–2.23] for DZ mixed sex twins. From this analysis, we suggest that twinning is not related to atypical laterality.

## Main analysis

The first step of analysis involved testing whether the laterality measures predicted whether or not a child was diagnosed with DLD using multilevel modelling. In the first model, the fTCD laterality index was used as a continuous measure, and in the second, a categoric measure was used, with children grouped as left, bilateral or right lateralised based on whether confidence intervals for the LI crossed with zero. Since we had clear a priori expectations that any group differences would involve reduced laterality in the DLD group, we did not correct for multiple testing. Contrary to hypothesis, there was no between-groups difference in fTCD LI, with the group factor being non-significant, $t(242.63) = 1.32$, $p = .190$. See Table 4 for marginal means and associated 95% CIs.

Following *Whitehouse & Bishop (2008)*, we also assessed via multilevel modelling whether mean blood flow in the left and right MCAs during the period of interest relative to the mean of the baseline was significantly different in the two groups. As can be seen in the grand average plots shown in Fig. 3, the time course of the changes in blood flow was very similar in both groups. Blood flow peaks bilaterally at the start of the trial, and it is only when flow returns to baseline levels in both MCAs that the left-sided bias emerges, with flow in the right MCA dropping below that in the left MCA.

There was no effect of group (TD or DLD) on mean percent change in flow in the left MCA, $t(258.28) = 1.94$, $p = .053$, or the right MCA, $t(260.97) = 1.72$, $p = .086$,

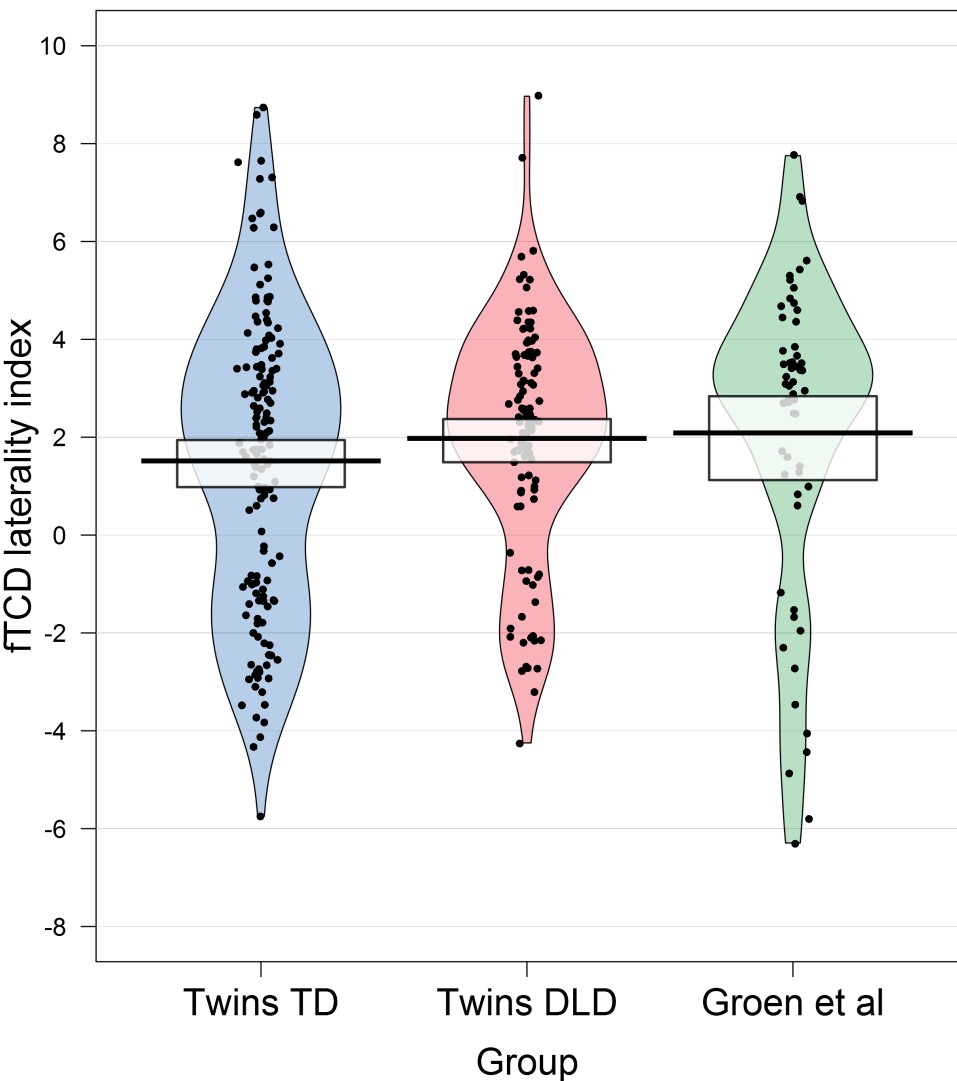

**Figure 2  Pirate plot showing fTCD laterality indices (LIs) for the twins in the current study.** For comparison, we also show LIs for children using the same fTCD task reported by *Groen et al. (2012)*. The twins are split as a function of group (TD or DLD), and all data points are shown with smoothed densities indicating the distributions in each sample. The central tendency is the mean and the intervals are Bayesian 95% Highest Density Intervals.

during the period of interest. See Table 4 for marginal means and associated 95% CIs. The non-significant trend for slightly lower flow in the TD group was likely driven by a greater range in blood flow in the talk phase relative to the baseline for the TD children (left: −15.80 to +9.00%; right: −18.40 to +10.00%), compared to those with DLD (left: −9.80 to +9.00%; right: −8.90 to +6.00%). Nonetheless, the overall impression of this analysis is of no significant between-group differences in cerebral blood flow during the language task.

**Table 4  Marginal means and 95% CIs for LI and mean % change in blood flow in the left and right MCAs during the period of interest compared to the baseline.**

|  | Marginal mean | Lower 95% CI | Upper 95% CI |
|---|---|---|---|
| TD: LI | 1.52 | 1.07 | 1.96 |
| DLD: LI | 1.98 | 1.44 | 2.51 |
| TD: Left flow, % change | −0.84 | −1.61 | −0.07 |
| DLD: Left flow, % change | 0.24 | −0.67 | 1.14 |
| TD: Right flow, % change | −2.14 | −2.88 | −1.39 |
| DLD: Right flow, % change | −1.19 | −2.07 | −0.32 |

Next, we specified a multinomial model testing for categoric differences in laterality in those with and without DLD. There was an effect of group on right compared to left laterality, $p$MCMC $= .007$, although this went in the opposite direction to that hypothesised, with an over-representation of TD children showing right lateralised language. There was no effect of group on bilateral compared to left laterality, $p$MCMC $= .503$. The predicted odds ratio (95% credibility intervals) of a TD child compared to a child with DLD being right rather than left lateralised was 3.42 [1.20, 8.90], and for being bilateral rather than left lateralised was 1.33 [0.57, 2.85].

We then moved to look at handedness, testing the hypothesis that the DLD group was less right-handed than the TD children. We ran inflated beta regression models with the adapted EHI and QHP as dependent variable, running each model in two replication samples, with random allocation of one twin to one sample and the other twin to the other. With logit-transformed handedness scores on the adapted EHI as dependent variable, the coefficients for group were non-significant in sample 1, $\beta = .02$, $t(131) = .08$, $p = .939$, and sample 2, $\beta = −.07$, $t(132) = −.31$, $p = .754$. This was the same for the regressions predicting logit-transformed QHP scores in sample 1, $\beta = .33$, $t(131) = 1.38$, $p = .171$, and sample 2, $\beta = −.24$, $t(132) = -.31$, $p = .284$. Note that sex and age showed no relationship with handedness, so these were not incorporated in any models.

We also tested the hypothesis that children with DLD were less likely in the QHP task to reach across the midline with the right hand to a spatial position on the left side of the body, indicating weaker hand preference, as previously reported (*Bishop, 2005*; *Hill & Bishop, 1998*). A multilevel model was run with twin pair membership as a random effect, group as a between factor and spatial position as a within factor. There was a large main effect of spatial position, $t(1,687.20) = 12.74$, $p < .001$, with individuals being less likely to reach across the midline with the right hand to a spatial position on the left. However, there was no main effect of group, $t(1,836.70) = 1.61$, $p = .108$, and contrary to previous reports, no interaction between DLD status and spatial position, $t(1,687.20) = 0.89$, $p = .376$, indicating no between-groups differences in strength of hand preference. See Fig. 4 for a plot of the probability of right hand reaches to each spatial location by group.

For the last part of the main analysis, we evaluated the hypothesis that inconsistency of left hemisphere dominance was associated with DLD. Firstly, the TD and DLD groups were divided into subgroups based on the combination of tasks on which a child showed

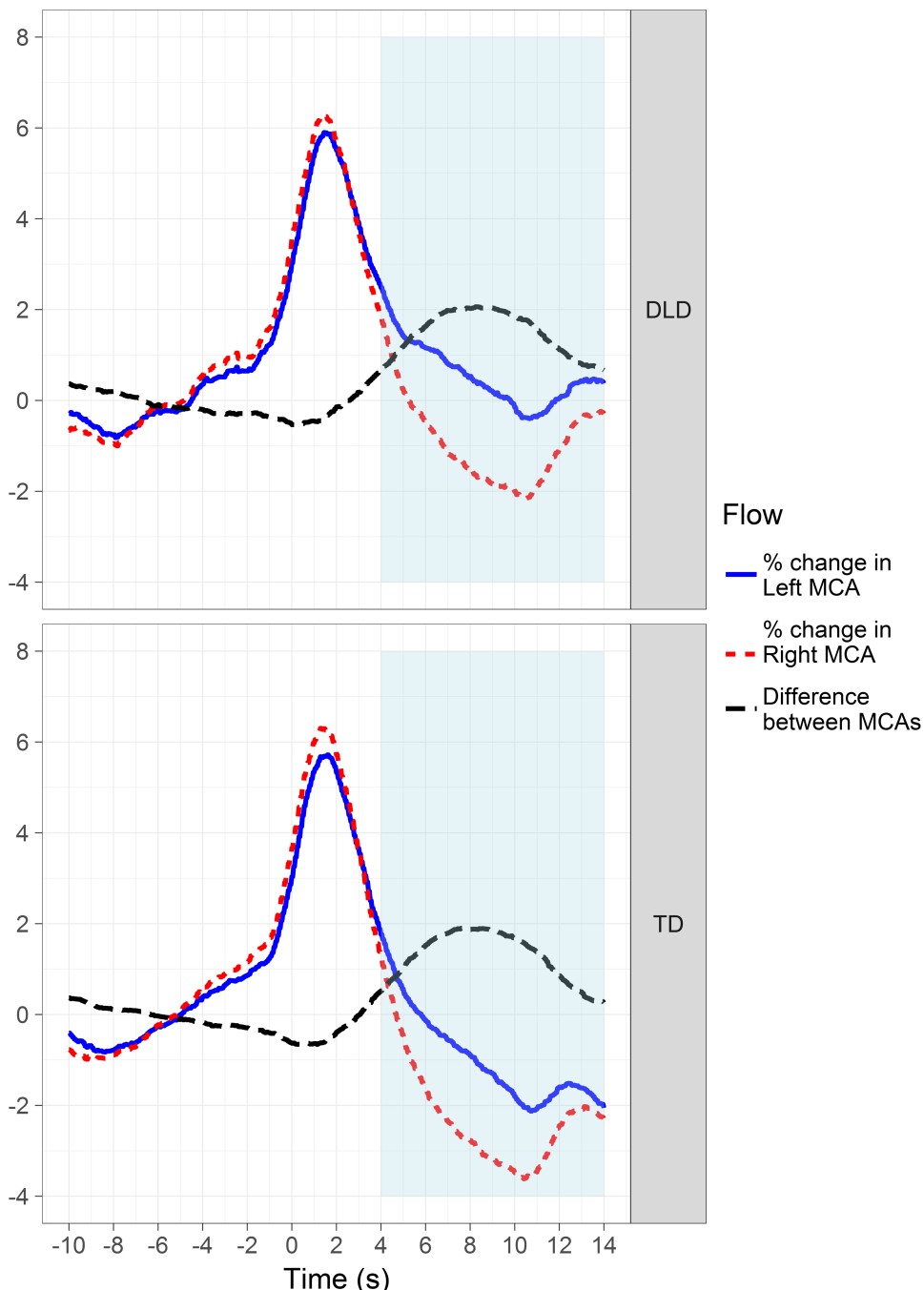

**Figure 3** **Plots showing the grand average curves for blood flow in the left and right MCAs for both groups.** The blue and red lines indicate blood flow in the two MCAs minus the mean baseline value, which is 100 following normalization and baseline-correction. Thus, a positive value indicates a percent increase above the mean of the baseline, and vice versa. The black line indicates the mean difference between flow in the two arteries, and therefore represents the lateralised response. The light blue area shows the period of interest during which language-related activity is measured.

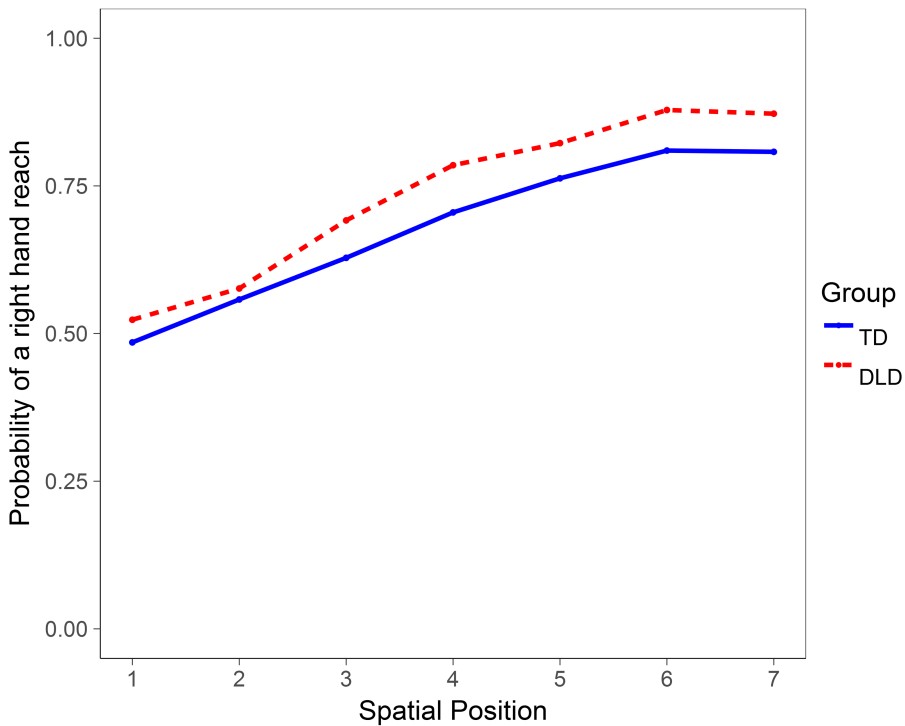

**Figure 4** **Plot showing slopes for each group reflecting the probability of making a right hand reach to each of seven spatial positions in the QHP task.** Position 4 marks the midline. Positions 1–3 are to the left of the participant and positions 5–7 are to the right, each placed at regular intervals of 30 degrees.

**Table 5** **Children are grouped based on evidence of left hemisphere dominance across the three laterality measures.** For each measure, 1 codes for left hemisphere dominant (i.e., left-lateralised language in fTCD and right-handedness in the EHI and QHP). In fTCD, 0 codes for bilateral or right lateralised language; in the handedness measures, 0 indicates that less than half of responses were right-handed.

| fTCD | EHI | QHP | TD, *n* boys | DLD, *n* boys | TD, *n* girls | DLD, *n* girls | TD, *n* children | DLD, *n* children |
|------|-----|-----|--------------|---------------|---------------|----------------|------------------|-------------------|
| 1 | 1 | 1 | 22 | 43 | 48 | 15 | 70 | 58 |
| 1 | 0 | 1 | 2 | 3 | 0 | 0 | 2 | 3 |
| 1 | 1 | 0 | 11 | 4 | 4 | 7 | 15 | 11 |
| 1 | 0 | 0 | 3 | 2 | 6 | 4 | 9 | 6 |
| 0 | 1 | 1 | 13 | 13 | 26 | 9 | 39 | 22 |
| 0 | 0 | 1 | 2 | 1 | 1 | 2 | 3 | 3 |
| 0 | 1 | 0 | 2 | 2 | 8 | 0 | 10 | 2 |
| 0 | 0 | 0 | 1 | 1 | 7 | 1 | 8 | 2 |

evidence of left hemisphere dominance; see Table 5 for the number of children falling into each group.

Then we tested whether there was a between-groups difference in the number of children falling into the 1,1,1 category versus any other category. For this purpose, we used a multilevel logistic regression, with group (TD or DLD) as a fixed effect and twin pair membership as a random effect. DLD showed no relationship with consistency of left hemisphere dominance, $z = 1.48$, $p = .139$, with this trait as frequent in the DLD group

as in the TD children. The predicted odds (95% CIs) for a child with DLD compared to a TD child showing consistent left hemisphere dominance was 1.48 [0.89, 2.52].

Since the overall percentage of children showing consistent left hemisphere dominance on the laterality measures was only around 50%, we checked whether relationships existed between the variables at the sample level, as lack of a relationship would complicate interpretation of that preceding analysis. For this purpose, we used multilevel models, with twin pair membership as a random effect. In the first model, we verified that the handedness measures were related. As expected, handedness measured by the adapted EHI was a significant predictor of quantitative handedness (QHP), $p < .001$. The measures shared a moderate amount of variance over and above the effect of twin pair membership, pseudo $R^2 = .21$. On the other hand, in two similar multilevel models with fTCD LI as dependent variable and one handedness measure as a fixed effect in each, the adapted EHI was not a significant predictor of LI, $p = .878$, and nor was the QHP, $p = .893$. Therefore, manual and language laterality were not related in our sample, though the separate measures of manual laterality were. With this in mind, we checked whether individual departures from the group pattern of right-handedness were a predictor of DLD in a second multilevel logistic regression. An individual was coded as right-handed (1) if they were right-handed on both measures, and coded as 0 otherwise. The fixed effect of group (TD or DLD) was also non-significant in this model, $z = .86$, $p = .388$. The predicted odds (95% CIs) for a child with DLD compared to a TD child showing consistent right-handedness was 1.28 [0.73, 2.29].

In the second step of analysis, we tested the relationship between language and laterality from the opposite direction, using language measures as predictors, aiming to replicate the effects reported by *Groen et al. (2012)*. We computed Spearman's correlations between each language measure and the fTCD laterality index in the two replication samples. However, no uncorrected correlation replicated across samples, and no correlation within a sample survived a Holm-Bonferroni correction. This was also the case when correlations were computed using absolute laterality indices, i.e., when testing the relationship between the strength of laterality rather than the laterality index per se, which is a combined measure of strength and direction.

Some of the children recruited into the study (though excluded in the main analysis) were reported by their parent(s) as having autism ($n = 12$). For completeness of reporting, we provide fTCD laterality results for this group below. Useable fTCD data were collected from 10 children with ASD, all of whom had an IQ above 70 (M = 100.18, SD = 12.69). These children showed relatively low left lateralisation for language function when assessed using fTCD, both in terms of the quantitative LI (M = 1.18, SD = 3.03), and in terms of categoric laterality (five were left lateralised, two showed bilateral language, and three were right lateralised). Two of the children were girls, both of whom were left lateralised. This analysis should not be given undue weight given the small sample, and it should be borne in mind that autism diagnoses were reported by parents, and were not confirmed in the course of this study using standardized clinical instruments.

## DISCUSSION

The present study evaluated whether reduced laterality for language was more common among those with developmental language problems. In our sample of 263 twins, we did not find any evidence for increased prevalence of atypical laterality in those with developmental language disorder. On the contrary, the fact that so many of the typically developing children in the sample were right lateralised for language when assessed using fTCD (19.90%) indicates that atypical laterality is not inconsistent with the development of typical language skills.

This study filled the need expressed by a recent review of the neuroimaging literature on childhood language disorder (*Mayes, Reilly & Morgan, 2015*) for a large confirmatory study of whether patterns of cerebral lateralisation are disrupted in language disorder. *Mayes, Reilly & Morgan (2015)* indicated that several exploratory studies provided limited evidence of neural differences, including laterality differences, in samples of language-impaired children, however methodological heterogeneity and variability in language phenotyping made it difficult to evaluate the strength of the evidence. The small sample sizes (no study included in the review exceeded 36 participants) and possible publication bias are also obvious problems thought these went unmentioned by the reviewers. Based on the large sample size of our present study, we suggest that previous reports of reduced left hemispheric dominance for language among those with language problems (e.g., *Illingworth & Bishop, 2009*; *Whitehouse & Bishop, 2008*) are likely to have been false positives. This result speaks to the chronic problem of low power that afflicts the neuroscience literature; *Button et al. (2013)* calculated a median statistical power of only 21% across 730 neuroscience studies incorporated in 48 meta-analyses published in 2011. Low powered studies are likely to fail to detect true effects where they exist, and the lower the average power of a study, the more the research literature will become disproportionately represented by false positives (*Sterne & Smith, 2001*). This is because, as power and therefore the number of true positives reduce, the number of false positives remains constant, since this is set by the *p*-value threshold. Low power should therefore generate low confidence that a positive finding in the research literature is a real effect, and this is before factoring in any further distortions created by publication bias and "flexible" analytical and/or reporting practices (*Button et al., 2013*). The moral of the present study is that the effects reported by small underpowered neuroscience studies should not be trusted until they are replicated in large well-powered studies.

The lack of a relationship between handedness and DLD in the present study also calls into question the mixed literature surrounding motor laterality and neurodevelopmental disorders. In particular, our failure to replicate a between-groups difference in the probability of reaching across the midline with the right hand indicates that this may not be a marker of compromised neurodevelopment relevant to language disorder, as previously suggested (*Bishop, 2005*; *Hill & Bishop, 1998*). We also failed to support the prediction of the left-brain bias model (*Bishop, 2013*) that reduced evidence of left hemisphere dominance would be found in the DLD group. This was the case even though around 50% of the children did not show evidence of left hemisphere dominance on all

three laterality variables. Indeed, there was a non-significant trend in the opposite direction hypothesised, with the TD group showing slightly less left hemisphere dominance across measures.

While the hypothesis was not supported, it may be premature to reject a possible role for a lack of consistent left hemisphere dominance in the development of language problems. Since this study used handedness measures and only one index of language lateralisation, we were not able to test a key prediction of the left-brain bias model, which stresses the importance of consistent left lateralisation across different language functions. By contrast, our assessment of left-hemisphere dominance was based on one language laterality measure and two measures of handedness. The relationship between handedness and language laterality measured by fTCD/fMRI is indirect at best (*Badzakova-Trajkov et al., 2010*; *Groen et al., 2013*; *Mazoyer et al., 2014*; *Somers et al., 2015a*), and in this respect, it is notable that the present study found that neither handedness measure predicted fTCD LI. This indicates that cerebral dominance for motor and language functions is likely to develop by largely independent processes, meaning that inconsistency between handedness and language laterality at the individual level need not reflect problems with hemispheric specialization. For a stronger test of the left-brain bias model, it would be necessary to identify individuals who do not show consistent left laterality across language tasks, which evoke moderately correlated patterns of lateralisation at the group level. In future work, we plan to test the prediction of the model that inconsistent lateralisation across language tasks will be associated with greater risk for DLD. Nevertheless, if the left-brain bias account were valid, we would expect to see at least a trend for reduced language lateralisation in the DLD group on the one measure we did have. The failure to observe such a trend in this large sample does weaken support for the model.

The absence of any relationship between lateralisation and language development cannot be attributed to a lack of individual variability in the data-set. On the contrary, considerable individual differences were evident in our sample—especially among the typically developing children, of whom nearly a fifth were right-lateralised during the animation description task used in our study. This finding remains in need of explanation. As one of our reviewers suggested, a possibility is that different children used different strategies to complete the animation description task. While it is wholly speculative, some children may have tended to construct mental images from the videos when describing them, which may have placed more demand on visuospatial cognition, which is typically right lateralised (*Badzakova-Trajkov et al., 2010*; *Jansen et al., 2004*).

There is a possibility that the null effect reported in this study is due to fTCD lacking the spatial resolution to pick up between-group differences if these are very fine-grained and focally located in the brain. All the same, if the distribution of language representation across the frontal and temporoparietal regions supplied by the middle cerebral artery does show widespread differences in laterality in DLD, we would expect to see the effects using fTCD. fTCD is highly sensitive to language-related activity, as confirmed by the high correlation reported between laterality indices produced using fTCD and fMRI for language tasks (*Deppe et al., 2000*; *Somers et al., 2011*). Furthermore, fTCD consistently shows a similar level of sensitivity to fMRI for the other commonly studied lateralised

cognitive process, visuospatial function (*Hattemer et al., 2011*; *Jansen et al., 2004*), and it also identifies categoric language dominance at a high level of agreement with the Wada test, in which the direct effect of anaesthetic injected into each MCA is observed on speech during neurosurgery (*Knake et al., 2003*; *Knecht et al., 1998*). We can therefore be confident in fTCD as a valid tool for measuring language lateralisation. We can also trust that the LIs reported in the present study reflected stable cerebral responses on a trial-by-trial basis given the high split-half reliability. Indeed, reproducibility of "gold-standard" fMRI measurements (*Adcock et al., 2003*; *Fernández et al., 2003*; *Jansen et al., 2006*; *Wilson et al., 2017*) is often lower than what is typically found for the fTCD LI (e.g., *Bishop, Watt & Papadatou-Pastou, 2009*).

## CONCLUSIONS

In a large sample of twins oversampled for language problems, the present study failed to find evidence for atypical laterality, either in terms of handedness or cerebral lateralisation for language, in those with DLD. Theories have proposed that disruption to the typical left hemisphere dominance for language may be a neurobiological correlate of language problems (e.g., *Annett, 2002*; *Bishop, 2013*; *Crow et al., 1998*), and empirical studies of very small samples have supported that view (*Illingworth & Bishop, 2009*; *Whitehouse & Bishop, 2008*). However, the present study did not replicate these findings, and we suggest that they are likely to have been false positives. In our large twin sample, fTCD testing revealed substantial individual variation in laterality, but the bias for left brain dominance for language showed no difference at the group level between those with and without DLD. We conclude, therefore, that reduced left hemisphere dominance is unlikely to be implicated in language disorder.

## ACKNOWLEDGEMENTS

We offer warmest thanks to the families who took part in the study, and school staff who helped facilitate assessment arrangements. The study would not have been possible without the hard work and dedication of a series of research assistants who conducted the assessments, often travelling all over the UK to do so: Eleanor Payne, Nicola Gratton, Georgina Holt, Annie Brookman, Elaine Gray, Louise Atkins, Holly Thornton and Sarah Morris. We also thank Paul A. Thompson for expert advice on statistical analysis and Margriet Groen for making data available for comparison with a previous study.

### Funding

This work was funded by Wellcome Trust Programme Grants no 082498/Z/07/Z and 082498/Z/07/C. The funders had no role in study design, data collection and analysis, decision to publish, or preparation of the manuscript.

## Grant Disclosures

The following grant information was disclosed by the authors:
Wellcome Trust Programme: 082498/Z/07/Z, 082498/Z/07/C.

## Competing Interests

Dorothy V.M. Bishop is an Academic Editor for PeerJ.

## Author Contributions

- Alexander C. Wilson analyzed the data, wrote the paper, prepared figures and/or tables, reviewed drafts of the paper.
- Dorothy V.M. Bishop conceived and designed the experiments, analyzed the data, wrote the paper, reviewed drafts of the paper, acquired funding.

## Human Ethics

The following information was supplied relating to ethical approvals (i.e., approving body and any reference numbers):

Ethical approval was obtained for the study in 2011 from the Berkshire NHS Research Ethics Committee. Approval number: Reference 11/SC/0096.

## Data Availability

Open Science Framework

https://osf.io/ksqrf/?view_only=54f013aaf65d45a5924748179538756d.

## Supplemental Information

Supplemental information for this article can be found online at http://dx.doi.org/10.7717/peerj.4217#supplemental-information.

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
