# Peer review of "Resounding failure to replicate links between developmental language disorder and cerebral lateralisation"

_PeerJ, doi:10.7717/peerj.4217_

## Round 0.1 · original submission · Minor Revisions

· Academic Editor

Minor Revisions

Both Reviewers indicated that this is a good study, which is likely to make a strong contribution to the literature in the field. However, both also suggested specific revisions which will improve the final published version, and I urge you to carefully address their recommendations in the revised version. I look forward to receiving your resubmission, and I thank you again for choosing PeerJ as a forum for your work.

Reviewer 1 ·

Basic reporting

The article is generally clear and well written, and the coverage of literature is appropriate. The tables and figures are clear. If anything, the data are over-analyzed, but this is understandable given that the results are largely negative with respect to the hypotheses under test—and with respect to previous claims. Some of the statistical techniques will be unfamiliar to many readers, but as far as I can tell they are appropriate.

I think the formula for computing LI should be given, along with more detail as to how their significance was tested for the assignment into laterality groups (binomial? T-test?).

The data have been made publically available.

Experimental design

One concern with the design is that the study was based on twins, and there might be some question as to whether the results can be generalized to nontwins. This is perhaps a relatively minor issue, although there is some evidence, again inconsistent, that twins may be slightly more likely to be left-handed than nontwins (as claimed, for example, by Annett). It might be worth citing the very large-scale study by Medland et al.. (2009, Neuropsychologia, 47, 330-337) showing effectively no difference in handedness between twins and nontwins, or between MZ and DZ twins. The idea that MZ twins of opposite handedness derive from mirror-splitting of the ovum can probably be dismissed, as the authors suggest, and it might be worth noting that the left-handed member of mirror pairs is typically left-dominant for language—just as left-handers in general are (e.g., Badzakova-Trajkov, et al., 2010. Neuropsychologia, 48, 3086-3093).

The authors cite McManus (1981) as evidence that handedness is unaffected by birth stress, although there are (or were) claims to the contrary, and there may be more than just birth stress that differentiates twins from nontwins.

It seems that the groups were not completely matched on IQ, and given the standard deviations I suspect the difference may be significant. Of course this partly reflects the language impairment in the DLD group, but I wondered whether they might be matched on nonverbal IQ.

It would also have been useful to know whether there were differences between the MZ and DZ twins. I realize that this will be the topic of another report, but I did wonder whether it might qualify the results reported here. The sex difference might also be a (minor) issue, and I think it is generally better to include possible confounding effects in the model rather than dismiss them if their effects are not significant.

These are actually fairly minor points, and the results are compellingly negative even if they are considered. Any minor deficiencies in design can be weighed against the fact that the samples were impressively large, and the analysis comprehensive. I might add though that previous reports of significant effects should not be attributed to lack of power, which has to do with Type II error, not Type I error.

Validity of the findings

In spite of some minor issues of design, the results are impressively negative with respect to the hypotheses being assessed. Acceptance of a null hypothesis (or rejection of the experimental hypothesis) is always an issue, given the logic of hypothesis testing, but here the p-values are impressively large.

I wondered if the finding of a slight right-hemisphere bias for the TD group might reflect different strategies during the fTCD sessions. Perhaps some of the participants were more likely than others to incorporate mental imagery while describing the cartoon action. I wonder if the results would have been different had a task without a visual component, such as word generation, been used. The authors do acknowledge that use of different measures of cerebral asymmetry might yield different results.

Comments for the author

This is a bold and commendable report, especially given that it seems to contradict some of your own previous findings.

Reviewer 2 ·

Basic reporting

The article meets the necessary standards - it is clear, with professional English used throughout.
The article does cite sufficient and relevant references although I note that one review that seems very relevant and complementary to this paper has not been cited - Mayes et al Dev Med Child Neurol. I think this was a 2014 or 2015 publication and am sure it focused on MRI evidence for or against the lateralisation hypothesis. It concluded that the field is a mess due to small numbers of participants, inconsistent MRI methodologies in terms of data acquisition and analysis methods etc. Whilst an MRI method was not applied in this paper, I would imagine that the findings of this review would serve to further strengthen the rationale for yet another study in this area (ie you address a number of the limitations of the field reviewed in their paper)?
The paper has a professional article structure and is self-contained with relevant results to the hypothesis posed.

Experimental design

Question is well defined. Strong participant numbers for this field. Rigorous investigation performed to a high standard.

Validity of the findings

No comment

Comments for the author

This is a careful body of work that builds on past related work by this leader of the field and team. I wonder if there is any room for inclusion of genetics discussion in this paper? Is Clyde Franks and team at Max Planck Institute in Nijmegen working in this space re genetics. Would be worth adding in a few lines on what has been found around hereditary links for language, lateralisation and handedness. Could this be an important missing variable in the debate in this field - ie we keep focusing on brain and behaviour but are genes missing here to make this a more sophisticated argument and get us into more specific genotyping of cases ...presumably this is a highly complex genetic relationship that will not be unravelled for many decades - but perhaps worth a mention as regards future direction...

---

## Round 0.2 · accepted · Accept

· Academic Editor

Accept

The revised version presents a strong case, and should be a valuable contribution to the existing literature in this discipline.